# Evaluation of Causal Inference Models to Access Heterogeneous Treatment Effect

## Abstract

Causal inference has gained popularity over the last years due to the ability to see through correlation and find causal relationship between covariates. There are a number of methods that were created to this end, but there is not a systematic benchmark between those methods, including the benefits and drawbacks of using each one of them. This research compares a number of those methods on how well they access the heterogeneous treatment effect using a variety of synthetically created data sets, divided between low-dimensional and high-dimensional covariates and increasing complexity between the covariates and the target. We compare the error between those method and discuss in which setting and premises each method is better suited.

## 1 Introduction

Over the last decades, causal inference has begun to gain importance within machine learning and data science discussions. Literature on the topic has grown considerably, especially after Judea Pearl came up with the do-calculus framework (Pearl, 1995), formalizing the theory of causality.

Even though there are a lot of benchmark papers comparing baseline and state-of-the-art causal inference models over frequently used data sets (namely IHDP (Hill, 2011) and NSW's jobs data set (LaLonde, 1986)), we have not been presented to date with an exhaustive algorithm comparison with data sets for which we know the true average treatment effect (ATE). Other authors have previously made surveys or comparisons of the most well known available methods for causal inference (Yao et al., 2021) (Shalit et al., 2017) (Alaa & Schaar, 2018), but none of them has used a range of different data generation processes nor compared the advantages and the restrictions of using each method.

We present a causal inference model benchmark using data sets from the Atlantic Causal Inference Conference (ACIC) 2019 data challenge (Gruber et al., 2019), which made available 6400 data sets generated by different data generation processes. In order to do the benchmark and further develop causal inference research and applications to the community, we developed an open source Python library called pycausal-explorer in which all models in the benchmark were implemented, as well as additional features that are useful for causal inference research.

## 2 Methodology

The main goal of this work is to benchmark different models for causal inference in a variety of settings, without prior knowledge of the data structure or the relation between the features, the treatment, and the target variables. We used models that represent different kinds of approaches: propensity score matching, IPTW, linear regression, k-nearest neighbors, extratrees embeddings (representing the matching algorithms), and x-learner (representing metalearners). The extratrees embedding and the x-learner experiments were divided into two different settings: one using the default hyperparameters of the models and another in which we did a step of hyper optimization to select the best parameters to use for the models.

We have also made a distinction among the algorithms we experimented with, classifying the most well known and longer used algorithms as baseline methods (propensity score matching, IPTW, linear regression, and

k-nearest neighbors); and the more recent algorithms shown to have good performance and generalization as state-of-the-art methods (x-learner and forest-model embedding).

Each model in the analysis was trained in every data set with standardized features whenever required by the model. We computed the mean average percentage error (MAPE) of the predicted average treatment effect (ATE) and grouped by the data generating processes (DGPs). The results are displayed in Tables 1, 2 and 3. In Table 2, we grouped the low dimensional data sets by modification used to generate the samples.

## 3  Data Sets

The data sets used in this study are from the ACIC 2019 Data Challenge. The competition featured 6400 data sets, from which half are qualified as low dimensional, ranging from 10 to 31 covariates, and the other half as high dimensional, ranging from 178 to 200 covariates. The objective is to estimate the average treatment effect (ATE) for each data set.

The data sets were synthesized from 64 different data generation processes (DGPs), which originated 100 different data sets each. The features were simulated or drawn from 7 different real data sources ranging from fields as diverse as cervical cancer, spam email, credit card defaults, student performance, right heart catheterization, epilepsy, and speed dating.

Additionally, a subset of the low dimensional data sets (2000 out of 3200 data sets) were created by four different processes:

1. The target is modeled as a linear regression of the variables (mod1)

2. There are complex iterations between the target and the features (mod2)

3. The target has a poor overlap between the treated and non treated, almost violating the positivity assumption (mod3)

4. There is heterogeneous treatment effect involved (mod4)

## 4  Neyman-Rubin Causal Model

In this study we will use the Neyman-Rubin framework for potential outcomes (Sekhon, 2008). The framework consists of a non-parametric model where each unit has two potential outcomes, one if the unit was treated and another if it was not. The treatment effect is calculated as the difference between such two potential outcomes. However, only one of the outcomes can be observed in practice, as the unit was either treated or not, which creates a conundrum called "the fundamental problem of causal inference".

Let $Y_i(1)$ be the potential outcome if the unit were to receive the treatment and $Y_i(0)$ otherwise. The individual treatment effect (ITE) is defined as:

$$\tau_i = Y_i(1) - Y_i(0) \tag{1}$$

Because $Y_i(1)$ and $Y_i(0)$ are never observed simultaneously, it is considered a missing data problem. Despite not being able to observe the ITE, we can make inferences about the average treatment effect (ATE). If the problem was designed as a randomized controlled trial, potential outcomes $Y_i(0), Y_i(1)$ would be independent of the treatment assignment T:

$$E[Yi(j)|T = 0] = E[Yi(j)|T = 1], j \in \{0, 1\} \tag{2}$$

This means that the group of the treated unit and the group of the untreated are comparable. Another way to see this is that the two groups are exchangeable, i.e. if the treated and the untreated were swapped it would return the same results. That is called the ignorability or the exchangeability assumption. The ATE would be defined as:

$$\tau = E[Y_i(1)|T = 1] - E[Y_i(0)|T = 0] = E[Y|T = 1] - E[Y|T = 0] \tag{3}$$

For an observational experiment, the independence assumption may not hold (as it is normally the case). Some assumptions are necessary in this case to get proper treatment effects, as described below.

### 4.1 Conditional exchangeability/unconfoundedness

In the cases where we have non-random treatment assignments (for instance, older people might be more susceptible to receive treatment than younger folks in certain clinical trials), we cannot use the ignorability assumption and thus the ATE formula. The unconfoundedness assumption is an extension of the ignorability assumption, where $Y_i(1)$, $Y_i(0)$ are independent of treatment T, given features X:

$$(Y(0), Y(1)) \perp\!\!\!\perp T|X \tag{4}$$

That means that there are no unobserved features that are confounders of the treatment assignment. Therefore, the ATE can be calculated as:

$$\tau = E_x\big[E[Y|T = 1, X] - E[Y|T = 0, X]\big] \tag{5}$$

### 4.2 Positivity

The positivity assumption determines that, in all subgroups, there is a probability greater than zero and lower than one to receive the treatment. If there is a subgroup that either received only treatment or control assignments, it would be not possible to calculate the treatment effect for that group.

$$0 < P(T = 1|X) < 1 \tag{6}$$

With this assumption, it is possible to evaluate equation 5 without dealing with undefined elements. As a side note, as the number of features grows, the likelihood of violating this assumption increases accordingly.

### 4.3 Stable unit-treatment value assumption (SUTVA)

This assumption deals with two ideas, the first being no interference: the potential outcome from one unit is not affected by the treatment assignment other units received. The second idea is no hidden variance: if a unit were to receive treatment, the outcome would be $Y = Y(1)$. That means that there are no possible different outcomes stemming from the same treatment.

## 5 Models

In this benchmark, we analyzed 6 algorithms as causal models, as described on Table 1. We investigated both baseline algorithms often present in the literature as well as methods which are the authors' take on more state-of-the-art causal models in machine learning. The latter are further discussed in the next 2 sections.

### 5.1 Randomized trees ensemble embedding

The randomized trees ensemble setting is inspired by the causal forest framework (Wager & Athey, 2018). The key concept for this setting is using a forest algorithm to model the relation between the covariates (without the treatment flag) and the outcome, and then matching treatment and control samples in the forest's latent embedding space. This setting is expected to overcome issues regarding confounder selection and heterogeneous treatment effects.

Table 1: Causal algorithms analyzed in this benchmark

| Model category | Name | Description |
|---|---|---|
| **Baseline models** | Propensity score matching (PSM) | Propensity score is modeled using logistic regression with balanced class weights; matching is performed on such score by a k-nearest neighbors model with 10 neighbors |
| | Inverse propensity of treatment weighting(IPTW) | Average treatment effect is computed as a weighted average with the same propensity scores used in the PSM setting described above |
| | Nearest neighbors | K-nearest neighbor models with 10 neighbors |
| | Linear models | Multiple linear regression without regularization for data sets with continuous targets |
| | | Logistic regression with balanced class weights for data sets with binary targets |
| **State-of-the-art models** | Randomized trees ensemble embedding | ExtraTrees embedding of the covariates followed by k-nearest neighbors matching with 10 neighbors |
| | X-learner | Meta learners using XGBoost as base model |

ExtraTrees models were chosen as the forest algorithms, because such models are expected to perform well in high dimensional settings with noisy covariates. The configuration is shown in Figure 1. In this benchmark, ExtraTrees with both default parameters and optimized parameters were included. As mentioned above, ExtraTrees models were trained to fit the outcome variable using the covariates. To make the models honest, as described in (Wager & Athey, 2018), we then proceed to use this model to encode the validation data set and then train two k-nearest neighbor models in the latent embedding space, one for the control set and another for the treatment set. Finally, counterfactuals and evaluation are computed for the test set.

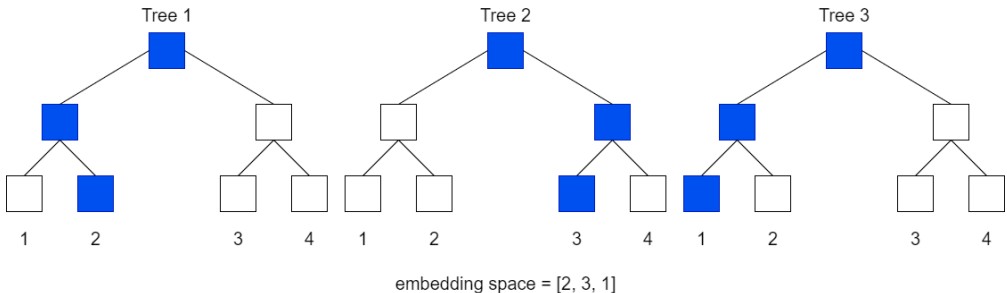

Figure 1: Forest-model embedding. The trained model yields an encoded representation of the training data on its leaves, forming an embedding which can be used in further steps of causal inference.

## 5.2 X-learner

The benchmark also included X-learners with the base model being a XGBboost model. Both models with the default parameters and with the hyperparameter tuning step were analyzed, as was the case for the randomized trees ensemble setting. The X-learner is implemented as described in (Künzel et al., 2019). The first step is to create two models (using the base model), one using only the data from the treated ($\mu_1$) and another using only the control data ($\mu_0$):

$$\mu_1 = E[Y(1)|X_{treated}] \tag{7}$$

$$\mu_0 = E[Y(0)|X_{control}] \tag{8}$$

The second step is to calculate the treatment effect from the treated and the treatment effect from the control:

$$D_1 = y_1 - \mu_0(X_{treated}) \tag{9}$$

$$D_0 = \mu_1(X_{control}) - y_0 \tag{10}$$

The third step is to create the models $\tau_0$ and $\tau_1$ based on the results from the last step:

$$\tau_0 = E[D_0|X_{control}] \tag{11}$$

$$\tau_1 = E[D_1|X_{treated}] \tag{12}$$

The last step to get the treatment effect is weighting the resulted models:

$$\tau(x) = g(x) * \tau_0(x) + (1 - g(x)) * \tau_1(x), g(x) \in [0,1] \tag{13}$$

The function $g(x)$ represents the propensity score, which is calculated as the output of a logistic regression trained using all data set.

## 6  Metrics and results evaluation

The main metric analysed in these data sets is the mean absolute percentage error (MAPE) of the predicted average treatment effect (ATE). This metric was chosen because ATE varies a lot in scale among data sets, so a relative and comparable measure was needed to do the benchmark.

The data sets were analyzed in 4 different settings:

1. Low dimensional data sets only;

2. Subset of the low dimensional data sets (2000 data sets out of 3200) grouped by 4 modifications used to generate the data sets. The modifications are: simple linear regression; target has a complex relationship with the features; treated and non-treated have a poor overlap regarding the features; heterogeneous treatment effect;

3. High dimensional data sets only;

4. High dimensional data sets grouped by 16 data generation scenarios.

## 7  Benchmark implementation

The pseudocode shown below implements the benchmark logic. It consists of a nested loop through all data sets and models, which are selected depending on the target type (regression models if the target is binary and classification models if the target is continuous).

Evaluation metrics are computed and stored after fitting the model, and the process is repeated for each data set.

```
for dataset in datasets:
        if dataset target is continuous:
                models = regression models
        else if dataset target is binary:
                models = classification models

        for model in models:
                fit model using data set
                compute evaluation metrics
                store evaluation metrics
```

Table 2: 95% confidence intervals for mean absolute percentage errors (MAPE) of predicted ATE for low dimensional data sets. Best performance (bold letters, darker blue) and second best performance (lighter blue) are indicated. Table show values for propensity score, IPTW, nearest neighbors and linear models.

| Target type | Propensity score | IPTW | Nearest neighbors | Linear models |
|---|---|---|---|---|
| Binary | **(44.59%, 44.69%)** | (86.93%, 87.13%) | (74.33%, 74.46%) | (60.47%, 60.56%) |
| Continuous | (20.28%, 20.32%) | (48.42%, 48.59%) | (55.56%, 55.64%) | **(15.56%, 15.63%)** |
| All | **(32.45%, 32.49%)** | (67.73%, 67.82%) | (64.97%, 65.02%) | (38.03%, 38.08%) |

Table 3: 95% confidence intervals for mean absolute percentage errors (MAPE) of predicted ATE for low dimensional data sets. Best performance (bold letters, darker blue) and second best performance (lighter blue) are indicated. Table show values for ExtraTrees Embedding and X-learner.

| Target type | ExtraTrees embedding (no hyperoptimization) | ExtraTrees embedding (with hyperoptimization) | Metalearners (no hyperoptimization) | Metalearners (with hyperoptimization) |
|---|---|---|---|---|
| Binary | (57.36%, 57.5%) | (58.76%, 58.9%) | **(29.7%, 29.76%)** | (32.68%, 32.76%) |
| Continuous | (23.1%, 23.15%) | (20.57%, 20.63%) | **(8.74%, 8.76%)** | (9.86%, 9.88%) |
| All | (40.25%, 40.31%) | (39.68%, 39.74%) | **(19.23%, 19.25%)** | (21.28%, 21.31%) |

We developed an open source Python library to help this work and forward the research of causal inference. All benchmark models and some sample data sets are available in the library. The package is available on pypi and can be installed via pip on Python 3.8 or higher. The source code can be found on GitHub[1] and it is under the MIT license.

# 8 Results

In this section, benchmark results are presented as 95% confidence intervals for the mean average percentage error (MAPE) of predicted ATE on different levels of aggregation. For more granular results (grouped by data generation process), please refer to Tables 10, 11, 12, and 13 in the Appendix.

## 8.1 Low dimensional setting

A summary of ATE MAPE results can be found on Tables 2 and 3, and it is evident that, for most cases, best performance is achieved by either the regression models or the meta learners.

Also, 95% confidence intervals for ATE MAPE results grouped by data generation modification setting can be found on Tables 4 and 5.

---

[1]https://github.com/gotolino/pycausal-explorer

Table 4: 95% confidence intervals for mean absolute percentage errors (MAPE) of predicted ATE for low dimensional data sets grouped by data generation process mode as implemented by the ACIC 2019 co-author Susan Gruber (from mode 1 to 4). Best performance (bold letters, darker blue) and second best performance (lighter blue) are indicated. Table show values for propensity score, IPTW, nearest neighbors and linear models.

| Mode | Propensity score | IPTW | Nearest neighbors | Linear models |
|---|---|---|---|---|
| 1 | (28.4%, 28.6%) | (48.6%, 49.0%) | (71.3%, 71.7%) | **(20.3%, 20.5%)** |
| 2 | **(39.0%, 39.2%)** | (100.6%, 101.3%) | (88.6%, 89.2%) | (59.5%, 60.0%) |
| 3 | **(25.6%, 25.8%)** | (71.6%, 72.4%) | (44.2%, 44.4%) | (26.0%, 26.2%) |
| 4 | **(17.9%, 18.0%)** | (31.7%, 31.9%) | (43.4%, 43.6%) | (27.1%, 27.3%) |

Table 5: 95% confidence intervals for mean absolute percentage errors (MAPE) of predicted ATE for low dimensional data sets grouped by data generation process mode as implemented by the ACIC 2019 co-author Susan Gruber (from mode 1 to 4). Best performance (bold letters, darker blue) and second best performance (lighter blue) are indicated. Table show values for ExtraTrees Embedding and X-learner.

| Mode | ExtraTrees embedding (no hyperoptimization) | ExtraTrees embedding (with hyperoptimization) | Metalearners (no hyperoptimization) | Metalearners (with hyperoptimization) |
|---|---|---|---|---|
| 1 | (30.2%, 30.4%) | (29.8%, 30.1%) | (16.8%, 16.9%) | **(14.1%, 14.2%)** |
| 2 | (64.5%, 65.0%) | (44.1%, 44.5%) | **(18.4%, 18.6%)** | (27.1%, 27.4%) |
| 3 | (24.4%, 24.5%) | (33.1%, 33.4%) | (16.0%, 16.2%) | **(15.8%, 16.0%)** |
| 4 | (17.1%, 17.2%) | (37.2%, 37.6%) | **(10.5%, 10.6%)** | (13.6%, 13.7%) |

Table 6: 95% confidence intervals for mean absolute percentage errors (MAPE) of predicted ATE for high dimensional data sets. Best performance (bold letters, darker blue) and second best performance (lighter blue) are indicated. Table show values for propensity score, IPTW, nearest neighbors and linear models.

| Target type | Propensity score | IPTW | Nearest neighbors | Linear models |
|---|---|---|---|---|
| Binary | **(16.89%, 16.93%)** | (32.28%, 32.35%) | (59.42%, 59.47%) | (99.22%, 99.22%) |
| Continuous | (37.43%, 37.65%) | (52.51%, 52.78%) | (57.34%, 57.45%) | **(13.83%, 13.94%)** |
| All | **(27.18%, 27.26%)** | (42.43%, 42.53%) | (58.4%, 58.44%) | (56.52%, 56.59%) |

## 8.2 High dimensional setting

A summary of ATE MAPE results can be found on Tables 6 and7, and again best performance is achieved by either the regression models or the meta learners in most cases.

The 95% confidence intervals for ATE MAPE results grouped by data generation scenario setting can be found on Tables 8 and 9.

## 9 Discussion

Benchmark results clearly show that x-learners had the lowest ATE MAPE in most of the settings, along with other findings. The main takeaways are described below.

- X-learners were the best performers across the board. It is probably due to the fact that such models are more robust to noisy covariates that might introduce bias to the ATE estimation;

- Propensity score matching (PSM) was the best baseline classification method for low and high dimensional settings alike, which is probably due to it being more robust to confounding and heterogeneous treatment effects;

- Linear regression was the best baseline regression method for both low and high dimensional settings, but specially in the high dimensional setting, which reflects the linear nature of most data generation processes for data sets with continuous outcomes;

Table 7: 95% confidence intervals for mean absolute percentage errors (MAPE) of predicted ATE for high dimensional data sets. Best performance (bold letters, darker blue) and second best performance (lighter blue) are indicated. Table show values for ExtraTrees Embedding and X-learner.

| Target type | ExtraTrees embedding (no hyperoptimization) | ExtraTrees embedding (with hyperoptimization) | Metalearners (no hyperoptimization) | Metalearners (with hyperoptimization) |
|---|---|---|---|---|
| Binary | (18.36%, 18.39%) | (20.54%, 20.6%) | **(10.0%, 10.03%)** | (12.51%, 12.54%) |
| Continuous | (37.73%, 37.91%) | (35.95%, 36.13%) | (17.0%, 17.09%) | (13.9%, 13.95%) |
| All | (28.07%, 28.13%) | (28.27%, 28.34%) | (13.51%, 13.55%) | **(13.22%, 13.24%)** |

Table 8: 95% confidence intervals for mean absolute percentage errors (MAPE) of predicted ATE for high dimensional data sets grouped by data generation process scenario as implemented by the ACIC 2019 co-author Geneviève LeFebvre [4] (from scenario 1 to 16). Best performance (bold letters, darker blue) and second best performance (lighter blue) are indicated. Table show values for propensity score, IPTW, nearest neighbors and linear models.

| Scenario | Propensity score | IPTW | Nearest neighbors | Linear models |
|---|---|---|---|---|
| 1 | (112.3%, 115.6%) | (198.3%, 204.0%) | (102.7%, 105.8%) | **(54.5%, 56.2%)** |
| 2 | (19.4%, 19.9%) | (29.6%, 30.4%) | (41.6%, 42.2%) | **(9.6%, 9.9%)** |
| 3 | (21.2%, 21.8%) | (36.2%, 37.3%) | (65.8%, 66.5%) | **(8.0%, 8.2%)** |
| 4 | (6.7%, 6.8%) | (9.2%, 9.5%) | (43.1%, 43.3%) | **(2.7%, 2.8%)** |
| 5 | (93.8%, 105.9%) | (108.9%, 119.8%) | (81.5%, 84.2%) | **(59.8%, 66.4%)** |
| 6 | (41.1%, 42.9%) | (57.3%, 60.4%) | (48.8%, 50.4%) | **(24.3%, 25.3%)** |
| 7 | (37.0%, 38.0%) | (40.6%, 42.3%) | (25.6%, 26.5%) | **(6.7%, 6.9%)** |
| 8 | (4.3%, 4.4%) | (6.2%, 6.4%) | (42.6%, 42.8%) | **(2.7%, 2.8%)** |
| 9 | (7.9%, 8.1%) | (10.7%, 11.0%) | (52.5%, 52.8%) | **(1.4%, 1.4%)** |
| 10 | (2.0%, 2.1%) | (4.3%, 4.4%) | (64.0%, 64.2%) | **(1.6%, 1.6%)** |
| 11 | (73.2%, 73.6%) | (75.5%, 76.3%) | (20.4%, 20.9%) | **(1.8%, 1.9%)** |
| 12 | (26.5%, 29.2%) | (39.0%, 45.4%) | (72.7%, 75.1%) | **(11.6%, 12.5%)** |
| 13 | (9.7%, 10.0%) | (18.8%, 19.3%) | (43.0%, 43.5%) | **(4.7%, 4.9%)** |
| 14 | (18.6%, 19.0%) | (22.0%, 22.7%) | (58.1%, 58.5%) | **(4.3%, 4.4%)** |
| 15 | (76.2%, 78.4%) | (105.6%, 108.9%) | (111.1%, 113.3%) | **(17.8%, 18.4%)** |
| 16 | (37.2%, 38.3%) | (61.2%, 63.2%) | (36.2%, 37.2%) | **(4.6%, 4.7%)** |

Table 9: 95% confidence intervals for mean absolute percentage errors (MAPE) of predicted ATE for high dimensional data sets grouped by data generation process scenario as implemented by the ACIC 2019 co-author Geneviève LeFebvre [4] (from scenario 1 to 16). Best performance (bold letters, darker blue) and second best performance (lighter blue) are indicated. Table show values for ExtraTrees Embedding and X-learner.

| Scenario | ExtraTrees embedding (no hyperoptimization) | ExtraTrees embedding (with hyperoptimization) | Metalearners (no hyperoptimization) | Metalearners (with hyperoptimization) |
|---|---|---|---|---|
| 1 | (116.0%, 119.2%) | (135.0%, 139.5%) | (61.6%, 63.2%) | **(61.5%, 63.0%)** |
| 2 | (21.9%, 22.4%) | (19.4%, 19.9%) | (11.4%, 11.7%) | **(8.2%, 8.4%)** |
| 3 | (28.9%, 29.6%) | (28.1%, 28.8%) | **(14.9%, 15.2%)** | (16.5%, 16.9%) |
| 4 | (6.1%, 6.3%) | (6.9%, 7.1%) | (3.2%, 3.3%) | **(2.5%, 2.6%)** |
| 5 | (73.8%, 81.1%) | (67.2%, 73.6%) | (57.2%, 61.6%) | **(35.0%, 36.0%)** |
| 6 | (43.9%, 45.6%) | (55.7%, 59.7%) | (32.7%, 34.3%) | **(18.2%, 18.7%)** |
| 7 | (16.3%, 17.2%) | (13.1%, 14.0%) | **(8.8%, 9.1%)** | (14.6%, 14.8%) |
| 8 | (6.7%, 6.8%) | (7.5%, 7.7%) | (1.9%, 1.9%) | **(1.6%, 1.7%)** |
| 9 | (3.0%, 3.0%) | (2.6%, 2.7%) | **(1.2%, 1.2%)** | (1.2%, 1.2%) |
| 10 | (2.7%, 2.8%) | (3.0%, 3.1%) | **(1.1%, 1.2%)** | (1.1%, 1.2%) |
| 11 | (52.0%, 52.4%) | (39.0%, 39.3%) | (16.4%, 16.5%) | **(5.8%, 5.9%)** |
| 12 | (17.5%, 19.8%) | (17.2%, 19.5%) | **(6.7%, 7.0%)** | (6.8%, 7.0%) |
| 13 | (12.6%, 12.8%) | (11.7%, 12.0%) | (4.7%, 4.9%) | **(4.6%, 4.8%)** |
| 14 | (9.6%, 9.9%) | (8.2%, 8.5%) | **(3.1%, 3.2%)** | (3.1%, 3.2%) |
| 15 | (155.9%, 158.0%) | (128.0%, 130.3%) | (34.4%, 35.1%) | **(31.7%, 32.4%)** |
| 16 | (27.7%, 28.5%) | (22.2%, 22.8%) | (8.3%, 8.5%) | **(7.6%, 7.8%)** |

- IPTW did worse than PSM, probably because when propensity scores are not well modeled (due to lack of correlation between covariates and the treatment flag), simply matching samples by score tends to do better than using such scores as inverse weights for ATE estimation;

- Nearest neighbors on the covariates performed poorly overall, which might indicate such algorithms are more sensitive to the lack of confounder selection (i.e. they're probably suffering from collider bias, mediator bias and noise from irrelevant covariates);

- Nearest neighbors in the ExtraTrees latent embedding space do better than nearest neighbors directly on the covariates, probably because the randomized forest embedding attenuates the confounder selection issue and filters out the noise;

- Hyper optimization did not systematically improve ATE estimation for ExtraTrees and x-learners in the low dimensional setting, which indicates that a better prediction power in the target does not necessarily imply in better causal inference, as considering confounding yields better treatment effect predictions than finding correlation. In the high dimensional setting, however, x-learners with hyper optimization did a little better than the default ones, which might indicate hyper optimization can come in handy when dealing with high dimensionality.

Simpler models like linear regression and propensity score matching are good starting points to benchmark results, but is important to address the premises from the data generation process first. Does the data not violate the positivity assumption? Do we have a hint that the target has simple interactions with the features? Then a linear regression could be a good starting point. If it is believed that the target has a complex iteration with the features, maybe matching algorithms are better choices. If the data has a very strong heterogeneous treatment effect, meta learners like the x-learner can be a good solution.

## 10 Conclusion

There is no silver bullet that is going to perform best in all settings. X-learners have gotten the best results overall, but was not the best across all DGPs. Linear regression and propensity score matching have performed well across the settings, even though they are considered to be simpler in comparison to the other methods analysed. Also, hyper optimization has not been a game changer for ATE estimation. The combination of all these factors seems to indicate that a better prediction power in the outcome does not necessarily imply in better causal inference.

Also, no confounder selection was performed, which might have hindered the performance of models that are more prone to issues with noise and biases. Including confounder selection principles is an important next step to further investigate causal model performance, because accounting for confounding is more effective than modeling correlation between covariates and outcomes in order to improve prediction of treatment effects.

We also hope that the open source library made available can be useful in further developing the area of causal inference, not only in research but also in real-life applications, with support of the community to continue the evolution of the library.

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

# A   Tables with detailed information for each data set

Table 11: Mean absolute percentage errors (MAPE) for low dimensional data sets grouped by data generation process. Best performance (bold letters, darker blue) and second best performance (lighter blue) are indicated. Table show values for ExtraTrees Embedding and X-learner.

| DGPid | Target type | Number of rows | Number of covariates | ExtraTrees embedding (no hyperoptimization) | ExtraTrees embedding (with hyperoptimization) | Metalearners (no hyperoptimization) | Metalearners (with hyperoptimization) |
|---|---|---|---|---|---|---|---|
| 17 | binary | 668 | 10 | 49.5% | 68.6% | 34.1% | **31.9%** |
| 18 | binary | 668 | 10 | 52.4% | 43.2% | **27.3%** | 30.8% |
| 19 | binary | 668 | 10 | 44.8% | 55.0% | 31.9% | **25.2%** |
| 20 | binary | 668 | 10 | 31.5% | 35.4% | 21.2% | **17.3%** |
| 21 | binary | 500 | 22 | 37.7% | 40.8% | **21.7%** | 23.7% |
| 22 | binary | 500 | 22 | 90.8% | 92.5% | **60.9%** | 95.6% |
| 23 | binary | 500 | 22 | 36.0% | 40.6% | 22.1% | **20.3%** |
| 24 | binary | 500 | 22 | 47.6% | 46.9% | **26.2%** | 30.9% |
| 29 | continuous | 5735 | 26 | 39.7% | 25.5% | 9.1% | **10.6%** |
| 30 | continuous | 5735 | 26 | 75.7% | 37.0% | 5.8% | **5.7%** |
| 31 | continuous | 5735 | 26 | 27.5% | 30.3% | 18.5% | **17.7%** |
| 32 | continuous | 5735 | 26 | 20.2% | 57.4% | **11.8%** | 28.3% |
| 33 | binary | 5735 | 26 | 12.1% | 13.3% | 15.8% | **9.5%** |
| 34 | binary | 5735 | 26 | 164.8% | 111.7% | **39.9%** | 79.1% |
| 35 | binary | 5735 | 26 | 12.9% | 40.2% | **8.7%** | 10.5% |
| 36 | binary | 5735 | 26 | 19.7% | 80.3% | **10.8%** | 12.6% |
| 37 | continuous | 500 | 22 | 25.2% | 15.3% | **5.6%** | 6.3% |
| 38 | continuous | 500 | 22 | 11.2% | 9.9% | 6.0% | **5.6%** |
| 39 | continuous | 500 | 22 | 27.1% | 18.7% | 6.0% | **5.4%** |
| 40 | continuous | 500 | 22 | 11.4% | 9.1% | **2.3%** | 2.4% |
| 41 | binary | 500 | 22 | 136.2% | 82.1% | 44.5% | **39.5%** |
| 42 | binary | 500 | 22 | 36.9% | 38.7% | **21.9%** | 23.1% |
| 43 | binary | 500 | 22 | 84.6% | 61.9% | 49.0% | **35.9%** |
| 44 | binary | 500 | 22 | 61.3% | 90.0% | 39.6% | **37.8%** |
| 57 | continuous | 649 | 31 | 31.7% | 27.0% | 15.5% | **8.1%** |
| 58 | continuous | 649 | 31 | 16.4% | 15.4% | **9.4%** | 9.5% |
| 59 | continuous | 649 | 31 | 17.7% | 21.4% | **10.0%** | 13.0% |
| 60 | continuous | 649 | 31 | 8.4% | 8.0% | **4.2%** | 5.8% |
| 61 | continuous | 649 | 29 | 18.3% | 15.2% | **9.6%** | 10.8% |
| 62 | continuous | 649 | 29 | 14.4% | 14.1% | **10.1%** | 11.1% |
| 63 | continuous | 649 | 29 | 19.3% | 19.4% | **11.3%** | 13.1% |
| 64 | continuous | 649 | 29 | 5.9% | 5.9% | **4.5%** | **4.5%** |

Table 10: Mean absolute percentage errors (MAPE) for low dimensional data sets grouped by data generation process. Best performance (bold letters, darker blue) and second best performance (lighter blue) are indicated. Table show values for propensity score, IPTW, nearest neighbors and linear models.

| Target type | Number of rows | Number of covariates | Propensity score | IPTW | Nearest neighbors | Regression models |
|---|---|---|---|---|---|---|
| binary | 668 | 10 | 56.3% | 94.8% | 116.9% | **30.4%** |
| binary | 668 | 10 | 37.3% | 108.0% | 112.3% | **29.4%** |
| binary | 668 | 10 | 39.0% | 63.4% | **38.5%** | 48.2% |
| binary | 668 | 10 | 35.8% | 61.1% | **28.5%** | 41.5% |
| binary | 500 | 22 | **33.2%** | 60.5% | 57.8% | 100.0% |
| binary | 500 | 22 | 96.3% | 108.9% | **40.6%** | 100.0% |
| binary | 500 | 22 | **32.3%** | 70.7% | 81.3% | 100.0% |
| binary | 500 | 22 | **42.9%** | 74.0% | 71.0% | 100.0% |
| continuous | 5735 | 26 | 28.4% | 41.3% | 86.0% | **9.0%** |
| continuous | 5735 | 26 | **60.0%** | 134.1% | 96.5% | 120.1% |
| continuous | 5735 | 26 | 37.9% | 194.7% | 30.9% | **17.5%** |
| continuous | 5735 | 26 | 17.7% | 33.8% | 56.0% | **11.2%** |
| binary | 5735 | 26 | **15.6%** | 25.6% | 65.8% | 40.8% |
| binary | 5735 | 26 | **66.9%** | 202.7% | 191.9% | 128.6% |
| binary | 5735 | 26 | **15.8%** | 30.6% | 45.8% | 46.2% |
| binary | 5735 | 26 | **18.4%** | 34.4% | 57.6% | 72.1% |
| continuous | 500 | 22 | 14.2% | 44.5% | 89.4% | **3.3%** |
| continuous | 500 | 22 | 8.1% | 16.1% | 98.4% | **4.8%** |
| continuous | 500 | 22 | 20.8% | 49.1% | 103.0% | **7.1%** |
| continuous | 500 | 22 | 10.4% | 19.2% | 14.9% | **4.1%** |
| binary | 500 | 22 | 69.1% | 154.3% | 80.4% | **32.1%** |
| binary | 500 | 22 | 33.5% | 69.8% | 67.5% | **20.6%** |
| binary | 500 | 22 | 51.0% | 119.3% | 61.6% | **29.0%** |
| binary | 500 | 22 | 70.9% | 114.5% | 72.7% | **49.3%** |
| continuous | 649 | 31 | 25.5% | 50.3% | 59.1% | **14.1%** |
| continuous | 649 | 31 | 16.9% | 32.1% | 21.2% | **9.2%** |
| continuous | 649 | 31 | 18.0% | 35.7% | 50.9% | **9.0%** |
| continuous | 649 | 31 | 7.2% | 13.6% | 31.2% | **3.8%** |
| continuous | 649 | 29 | 16.6% | 31.9% | 29.8% | **7.6%** |
| continuous | 649 | 29 | 14.4% | 27.7% | 22.4% | **11.5%** |
| continuous | 649 | 29 | 17.8% | 35.7% | 55.5% | **9.9%** |
| continuous | 649 | 29 | 10.8% | 16.1% | 44.2% | **7.3%** |

Table 12: Mean absolute percentage errors (MAPE) for high dimensional data sets grouped by data generation process. Best performance (bold letters, darker blue) and second best performance (lighter blue) are indicated. Table show values for propensity score, IPTW, nearest neighbors and linear models.

| DGPid | Target type | Number of rows | Number of covariates | Propensity score | IPTW | Nearest neighbors | Regression models |
|---|---|---|---|---|---|---|---|
| 1 | continuous | 1000 | 200 | 114.0% | 201.1% | 104.2% | **55.3%** |
| 2 | continuous | 1000 | 200 | 19.7% | 30.0% | 41.9% | **9.7%** |
| 3 | continuous | 1000 | 200 | 21.5% | 36.7% | 66.1% | **8.1%** |
| 4 | continuous | 2000 | 200 | 6.7% | 9.3% | 43.2% | **2.7%** |
| 5 | continuous | 2000 | 200 | 99.8% | 114.3% | 82.8% | **63.1%** |
| 6 | continuous | 1000 | 200 | 42.0% | 58.8% | 49.6% | **24.8%** |
| 7 | continuous | 2000 | 200 | 37.5% | 41.4% | 26.1% | **6.8%** |
| 8 | continuous | 1000 | 200 | 4.3% | 6.3% | 42.7% | **2.7%** |
| 9 | continuous | 2000 | 200 | 8.0% | 10.8% | 52.6% | **1.4%** |
| 10 | continuous | 2000 | 200 | 2.0% | 4.4% | 64.1% | **1.6%** |
| 11 | continuous | 2000 | 200 | 73.4% | 75.9% | 20.6% | **1.9%** |
| 12 | continuous | 2000 | 200 | 27.8% | 42.2% | 73.9% | **12.1%** |
| 13 | continuous | 2000 | 200 | 9.8% | 19.1% | 43.2% | **4.8%** |
| 14 | continuous | 1000 | 200 | 18.8% | 22.3% | 58.3% | **4.3%** |
| 15 | continuous | 2000 | 200 | 77.3% | 107.3% | 112.2% | **18.1%** |
| 16 | continuous | 2000 | 200 | 37.8% | 62.2% | 36.7% | **4.7%** |
| 25 | binary | 1500 | 178 | **10.8%** | 19.3% | 60.3% | 98.1% |
| 26 | binary | 1500 | 178 | **26.7%** | 42.1% | 40.7% | 94.4% |
| 27 | binary | 2000 | 178 | **12.9%** | 25.1% | 61.6% | 98.6% |
| 28 | binary | 2000 | 178 | **20.3%** | 29.5% | 37.6% | 98.3% |
| 45 | binary | 2000 | 185 | **26.6%** | 39.3% | 59.1% | 99.8% |
| 46 | binary | 2000 | 185 | **17.1%** | 34.1% | 55.9% | 99.8% |
| 47 | binary | 2000 | 185 | **21.5%** | 40.8% | 56.8% | 99.9% |
| 48 | binary | 2000 | 185 | **15.5%** | 29.8% | 72.6% | 99.9% |
| 49 | binary | 2000 | 185 | **14.8%** | 29.7% | 65.4% | 99.9% |
| 50 | binary | 2000 | 185 | **13.1%** | 25.3% | 67.9% | 99.9% |
| 51 | binary | 2000 | 185 | **26.5%** | 43.9% | 71.4% | 99.8% |
| 52 | binary | 2000 | 185 | **21.2%** | 47.4% | 75.0% | 99.9% |
| 53 | binary | 2000 | 185 | **22.2%** | 67.2% | 35.5% | 100.0% |
| 54 | binary | 2000 | 185 | **8.9%** | 24.0% | 58.3% | 99.8% |
| 55 | binary | 2000 | 185 | **4.4%** | 5.9% | 70.7% | 99.9% |
| 56 | binary | 2000 | 185 | **8.1%** | 13.5% | 62.4% | 99.8% |

Table 13: Mean absolute percentage errors (MAPE) for high dimensional data sets grouped by data generation process. Best performance (bold letters, darker blue) and second best performance (lighter blue) are indicated. Table show values for ExtraTrees Embedding and X-learner.

| Target type | Number of rows | Number of covariates | ExtraTrees embedding (no hyperoptimization) | ExtraTrees embedding (with hyperoptimization) | Metalearners (no hyperoptimization) | Metalearners (with hyperoptimization) |
|---|---|---|---|---|---|---|
| continuous | 1000 | 200 | 117.6% | 137.3% | 62.4% | **62.2%** |
| continuous | 1000 | 200 | 22.1% | 19.7% | 11.5% | **8.3%** |
| continuous | 1000 | 200 | 29.3% | 28.4% | **15.1%** | 16.7% |
| continuous | 2000 | 200 | 6.2% | 7.0% | 3.3% | **2.5%** |
| continuous | 2000 | 200 | 77.5% | 70.4% | 59.4% | **35.5%** |
| continuous | 1000 | 200 | 44.7% | 57.7% | 33.5% | **18.4%** |
| continuous | 2000 | 200 | 16.7% | 13.5% | **9.0%** | 14.7% |
| continuous | 1000 | 200 | 6.7% | 7.6% | 1.9% | **1.7%** |
| continuous | 2000 | 200 | 3.0% | 2.6% | **1.2%** | **1.2%** |
| continuous | 2000 | 200 | 2.8% | 3.0% | **1.1%** | 1.2% |
| continuous | 2000 | 200 | 52.2% | 39.1% | 16.4% | **5.9%** |
| continuous | 2000 | 200 | 18.7% | 18.3% | **6.9%** | **6.9%** |
| continuous | 2000 | 200 | 12.7% | 11.9% | 4.8% | **4.7%** |
| continuous | 1000 | 200 | 9.8% | 8.3% | **3.2%** | **3.2%** |
| continuous | 2000 | 200 | 157.0% | 129.2% | 34.7% | **32.1%** |
| continuous | 2000 | 200 | 28.1% | 22.5% | 8.4% | **7.7%** |
| binary | 1500 | 178 | 15.1% | 12.9% | 7.5% | **6.3%** |
| binary | 1500 | 178 | 13.9% | 18.0% | **10.8%** | 12.6% |
| binary | 2000 | 178 | 14.1% | 27.7% | **7.0%** | 14.4% |
| binary | 2000 | 178 | 16.5% | 23.3% | **13.4%** | 21.1% |
| binary | 2000 | 185 | 20.4% | 23.7% | **12.3%** | 16.3% |
| binary | 2000 | 185 | 23.5% | 22.5% | **10.9%** | 18.0% |
| binary | 2000 | 185 | 22.2% | 18.3% | **12.8%** | 18.5% |
| binary | 2000 | 185 | 17.7% | 22.1% | **10.4%** | 11.4% |
| binary | 2000 | 185 | 22.3% | 17.9% | **7.6%** | 15.5% |
| binary | 2000 | 185 | 16.7% | 16.0% | **8.4%** | 14.1% |
| binary | 2000 | 185 | 21.5% | 25.7% | **13.0%** | 13.7% |
| binary | 2000 | 185 | 34.3% | 28.0% | 15.4% | **14.1%** |
| binary | 2000 | 185 | 31.0% | 49.5% | 18.3% | **13.7%** |
| binary | 2000 | 185 | 9.5% | 10.1% | 5.4% | **4.9%** |
| binary | 2000 | 185 | 4.2% | 6.5% | 2.2% | **1.8%** |
| binary | 2000 | 185 | 10.9% | 6.9% | 4.9% | **4.4%** |

