# OpenReview forum: "Evaluation of Causal Inference Models to Access Heterogeneous Treatment Effect"
_TMLR — Rejected by TMLR_

### Review · Reviewer_vjH2 · 2023-02-27

**Summary Of Contributions:**

This paper aims to compare some methods that estimate heterogeneous treatment effects. In particular, they consider scenarios with low-dimensional and high-dimensional covariates and different complexities between the covariates and the target and focus on propensity-based methods, direct methods, tree-based methods, and meta-learners. The conclusion is mixed as there is no silver bullet for all settings.

**Audience:**

No

**Broader Impact Concerns:**

Not seen.

**Claims And Evidence:**

No

**Requested Changes:**

See the comments in **Weaknesses** above.

**Strengths And Weaknesses:**

**Strengths**

1. This paper considers different scenarios, including low-dimensional and high-dimensional covariates and different complexities between the covariates and the target.
2. This paper reviews the detailed causal inference framework for heterogeneous treatment effects.

**Weaknesses**

1. As an evaluation paper, the evaluation metrics only use the mean absolute percentage error (MAPE), and there are no any additional evaluation criteria for different scenarios.
2. It is also very disappointing that the current paper only compares a very limited number of methods for estimating heterogeneous treatment effects. The baseline methods are not well-organized, and the state-of-the-art models only focus on two works published in 2018. I highly recommend the authors to check more recent works.
3. The conclusion is not informative, without any deep investigation of the performances under different models.
4. The writing and presentation are despairing with typos and grammatical mistakes here and there. Please do proofreading.

---

> ### Author Response · Authors · 2023-04-10
> **Review comment**
>
> Thanks for the review! We are working on the suggested improvements. We are working on adding more state of the art models in the benchmark with additional evaluation criteria, as well as a section explaining in more depth the benchmark models in the form of a review.

---

### Review · Reviewer_uy8f · 2023-03-02

**Summary Of Contributions:**

This paper develops a benchmark for evaluating causal inference methods. The authors evaluate a number of causal inference techniques using the datasets from the ACIC 2019 Data Challenge. The mean absolute percentage errors of different methods on different datasets are presented in the paper.

**Audience:**

No

**Broader Impact Concerns:**

No concerns.

**Claims And Evidence:**

Yes

**Requested Changes:**

1. Add more state-of-the-art causal inference models into the discussions.
2. Compare the experimental results with the ACIC 2019 Data Challenge.
3. Discuss whether the data generation satisfies the assumptions of a model and discuss how it affects the performance of the model.
4. Point out the difference between the potential outcome framework and Pearl’s structural equation model framework.


**Strengths And Weaknesses:**

Strength:
It is good to have a benchmark for evaluating causal inference methods.

Weakness:
1. Only two state-of-the-art models are evaluated in the paper. Causal inference has attracted more and more attention in recent years and many models have been proposed. As a benchmark, more state-of-the-art models are expected to be involved in the evaluation.

2. Since the data come from the ACIC 2019 Data Challenge, it will be interesting to compare the experimental results with the results from the ACIC 2019 Data Challenge.

3. Readers are interested to know the performance of different models under different assumptions. The golden rule of causal inference is that causal effects cannot be inferred purely from statistics. Thus, every causal inference model has made assumptions and different models may have different assumptions.  It is interesting to know the performance of different models when the data generation process satisfies or does not satisfy the assumptions made by the models. The paper mentions several common assumptions and groups the data based on the generation process. However, it is also expected to discuss the influences of those assumptions in the experiments, e.g., how well the assumptions of different models match the data generation process and how this matching affects the model performance.

4. It is well-known that there are two main frameworks for causal inference: the potential outcome framework and Pearl’s structural equation model framework. It is fine to only discuss the methods from the potential outcome framework, but the authors should explicitly mention that in the paper to avoid confusion.

---

> ### Author Response · Authors · 2023-04-10
> **Review comment**
>
> Thanks for the review! We are working on the suggested improvements. We are working on adding more state of the art models in the benchmark. We will also address in more detail the assumptions of the data generation processes and how models behave in such assumptions.

---

### Review · Reviewer_JLFv · 2023-03-16

**Summary Of Contributions:**

The authors compare a few popular methods of estimating average treatment effect in the causal inference literature. In particular, they look at 6400 data sets from the ACIC 2019 Data Challenge. They empirically evaluate the performance of the methods on the data sets and provide explanations for the results.

**Audience:**

No

**Claims And Evidence:**

No

**Requested Changes:**

1. The goal of the paper. When I started reading the paper, I thought the paper was about estimating heterogenous treatment effect. Towards the middle part of the paper, I realized the authors really want to focus on estimating the average treatment effect. It will be helpful if the authors can make this clearer in the title, abstract and introduction.
2. Citations and references. It will be helpful if the authors can cite more relevant papers. This is to (1) acknowledge the contribution of prior work, and (2) help readers who are not familiar with the field understand the context better. For example, in section 2, when the authors mention names of the methods, it will be helpful to cite relevant papers. In the first sentence of Section 4, it will be helpful to cite Neyman and Rubin’s older works for the potential outcome framework.
- Even better than this, the authors may include a related work section in the paper. There, the authors can talk about the papers that study the various methods (e.g. x-learner). They can also talk about previous works on the 2019 ACIC data sets. What have been done with the data sets? In addition, they can talk about what are some benchmark methods people use in the literature. How do people claim one method is better than the other?
3. Interpretations of the results. Following the results, the authors attempt to explain the results. However, many of the explanations/interpretations are not supported by evidence. It will be helpful if the authors can provide some evidence of their claim. For example, the authors claim that “Propensity score matching (PSM) was the best baseline classification method for low and high dimensional settings alike, which is probably due to it being more robust to confounding and heterogeneous treatment effects”. The data sets are synthetic data sets, thus it is possible to check whether PSM actually does better in more heterogeneous settings. The authors can, for example, do some data visualization, plotting the MAPE of each data set against some heterogeneity measure of the treatment effect of this data set. Then it will be clear whether PSM does do better in more heterogeneous settings.
4. Equation (2). It will be better if we use a or w for the treatment instead of j. j is usually reserved for index.
5. Below equation (6). I don’t think the following statement is 100% accurate “As a side note, as the number of features grows, the likelihood of violating this assumption increases accordingly.” In experimental settings, the assumption is always true. It will be helpful to emphasize that this is for observational studies, and this is not always true, but this phenomenon is common/we expect to see such a phenomenon.


**Strengths And Weaknesses:**

Strengths

1. The results are explained in a clear way.
2. The paper addresses an interesting and important question: empirically evaluating the performance of popular methods in the causal inference literature on a large, diverse set of data sets.

Weaknesses

See “Requested Changes” for more details.

---

> ### Author Response · Authors · 2023-04-10
> **Review comment**
>
> Thanks for the review! We are working on the suggested improvements. The paper addresses the implementation of models to estimate treatment effect in an heterogeneous treatment effect data generation process, we will make it clearer in the article. We will also add a section describing prior work in depth, explaining more the baseline methods.

---

### Review · Reviewer_fzfB · 2023-03-21

**Summary Of Contributions:**

The paper aims to fill the gap of non-existing systematic evaluations for estimating treatment effects in the presence of heterogeneity.
Both the research and practitioners community can benefit from such framework which can contribute in using causal inference methods in a more informed manner.

**Audience:**

Yes

**Broader Impact Concerns:**

No ethical concerns.

**Claims And Evidence:**

No

**Requested Changes:**

- Please review recent ML conferences and journals for papers related to the topic and update the text and experiments accordingly
- Please include synthetic controlled dataset  in your experiments
- In my opinion the whole text should be rewritten in a more structured way. Some suggestions improve introduction with relevant lit review, clearly state hypothesis tested (research questions) and takeaway messages from the benchmark at the end of the introduction. You could maybe get inspired by [1]





[1] Qin, Tian, Tian-Zuo Wang, and Zhi-Hua Zhou. "Budgeted heterogeneous treatment effect estimation." International Conference on Machine Learning. PMLR, 2021.

**Strengths And Weaknesses:**

Strengths:
- lack of comprehensive benchmark for measuring HTE is an important problem to tackle
- the authors suggest systematically exploring different experimental setups (dimensionality of covariates, complexity of relationships, complexity of estimators etc)

Weaknesses:
- although the text is easy to read, I don't find the paper self contained especially with regards to measuring causal effects. The introduction and background can be better written to improve the flow and presentation of the work.
- Literature review is lacking which hinders the experimental section as well
- The HTE is usually measured by PEHE, I wonder why the authors didn't use this metric?
- more advanced baselines are missing in the benchmark
- evaluating the baseline models on synthetic simulations will strengthen the presented conclusions/discussion

---

> ### Author Response · Authors · 2023-04-10
> **Review comment**
>
> Thanks for the review! We are working on the suggested improvements. We are working on adding more state of the art models in the benchmark, as well as a section explaining in more depth the benchmark models in the form of a review. The benchmark models were selected as representative of their types (matching algorithms, reweighting, tree-based and linear models). Did you miss any benchmark that should also be addressed here?

---

### Decision · Action_Editors · 2023-04-22

**Recommendation:** Reject

**Comment:**

This paper aims to build a benchmark to evaluate causal effect estimation algorithms. To this end, this paper creates synthetic data with varying covariate dimensions and evaluate the methods for heterogeneous treatment effect estimation.

While the motivation of this paper is very good, the proposed solution has some critical limitations. First, the survey of existing methods is far from complete and misses recent advanced methods. Second, many of the explanations/interpretations are not supported by evidence. Third, the assumptions under which causal inference could work are not well studied in the evaluation of the methods, for example, what if some assumptions are violated. Finally, there is only one evaluation metric, which makes the conclusion less convincing.  Therefore, I do not recommend acceptance of this paper.

**Audience:**

Yes

**Claims And Evidence:**

No.

The interpretation of results are not well supported by empirical evidence.

Evaluation metrics are limited; commonly used PEHE is not included.

The baseline methods do not include the most recent ones.